# Antiallergic Effects of *Callerya atropurpurea* Extract In Vitro and in an In Vivo Atopic Dermatitis Model

**DOI:** 10.3390/plants12040860

**Published:** 2023-02-14

**Authors:** Wooram Choi, Hwa Pyoung Lee, Philaxay Manilack, Veosavanh Saysavanh, Byoung-Hee Lee, Sarah Lee, Eunji Kim, Jae Youl Cho

**Affiliations:** 1Department of Integrative Biotechnology, Sungkyunkwan University, Suwon 16419, Republic of Korea; 2Department of Forestry, Ministry of Agriculture and Forestry, Vientiane P.O. Box 811, Laos; 3National Institute of Biological Resources, Environmental Research Complex, Incheon 22689, Republic of Korea; 4R&D Center, Yungjin Pharmaceutical Co., Ltd., Suwon 16229, Republic of Korea

**Keywords:** *Callerya atropurpurea*, allergy, atopic dermatitis, Lyn kinase

## Abstract

(1) Background: *Callerya atropurpurea* is found in Laos, Thailand, and Vietnam. Although the anti-inflammatory action of *C. atropurpurea* has been investigated, the functions of this plant in allergic responses are not understood. Here, we explored the antiallergic mechanism of *C. atropurpurea* ethanol extract (Ca-EE) using in vitro assays and an in vivo atopic model. (2) Methods: The constituents of Ca-EE were analyzed using GC/MS. Inhibition of lipoxygenase and β-hexosaminidase activity was examined, and the expression of inflammatory genes was measured by quantitative real-time PCR. The regulatory roles of Ca-EE in IgE/FcεRI signaling were examined by Western blotting. The DNCB-induced atopic dermatitis mouse model was performed with histological analysis. (3) Results: Ca-EE comprised cis-raphasatin, lupeol, some sugars, and fatty acids. In RBL-2H3 cells, treatment with Ca-EE significantly reduced the activities of lipoxygenase and β-hexosaminidase, as well as cytokine gene expression. IgE-mediated signaling was downregulated by blocking Lyn kinases. Moreover, Ca-EE effectively inhibited allergic symptoms in the DNCB-induced atopic dermatitis model without toxicity. (4) Conclusions: Ca-EE displayed antiallergic activities through regulating IgE/Lyn signaling in RBL-2H3 cells and a contact dermatitis model. These results indicate that Ca-EE could be effective for allergic disease treatment.

## 1. Introduction

Allergic disorders are type I hypersensitivity reactions, which are IgE-mediated immune responses against diverse antigens/allergens. This reaction is immediately provoked, and allergen/IgE-bound mast cells and basophils are essential components in atopy [1,2,3,4]. In immunological cells, IgE is crosslinked to FceRI, which activates signaling cascades. Src family kinases, Lyn and Fyn, are recruited to immunoreceptor tyrosine-based activation motifs (ITAM) in FcεRI β and γ chains, with sequential phosphorylation of tyrosine residues of the chains. Activated Lyn transduces the signals to Syk kinase, which results in a linker for the activation of T cells (LAT). The reaction is required for the activation of phospholipase C (PLC)γ, which modulates Ca^2+^ mobilization and protein kinase C (PKC) activity. These signals lead to mast cell granulation or transcription factor activation for inflammatory cytokines such as IL-1β, IL-4, IL-5, and TGF-β [3,5,6,7]. IL-1β plays critical roles in inflammation, contact hypersensitivity, and atopic dermatitis. Allergen-mediated inflammasome induces the secretion of IL-1β, which exacerbates symptoms [8]. IL-4 regulates T-cell development and IgE production, and IL-5 is considered a hallmark of atopic diseases [9,10]. TGF-β modulates leukocyte chemotaxis and Th17 cell differentiation [11]. Under uncertain conditions to overproduce IL-2, IL-10, and TGF-β from T cells and macrophages, the acute phase of allergic inflammation can be switched to chronic allergy and atopic dermatitis conditions, under which T-cell proliferation is increased, IL-4, IL-5, and GM-CSF are overproduced, and IgE and IgM are highly produced [12,13]. 

Mast cells or basophils also produce histamine, the main contributor of allergic responses, which are bound to histamine receptors H1, H2, H3, or H4 [14,15]. Histamine mediates inflammatory cytokine secretion and promotes vascular expansion and tissue changes [16]. β-Hexosaminidases are necessary for histamine release. The enzymes are simultaneously degranulated with histamine in mast cells, a known biomarker of allergic reaction [17]. Lipoxygenases are also involved in histamine release through arachidonic acid metabolism [18].

Atopic dermatitis (AD) is one of the chronic inflammatory and IgE-related allergic diseases accompanying edema, erythema, pruritus, or eczema [1,19,20,21]. The first-line treatment for AD is represented by corticosteroids such as dexamethasone or prednisolone, which downregulate inflammatory transcriptional factors for proinflammatory cytokines or mediators [22,23]. However, an allergic reaction to corticosteroids has been reported [23], and prolonged administration of corticosteroids can bring adverse effects including gastrointestinal side-effects, hypertension, skeletal muscle atrophy, or glucose intolerance [24,25,26]. Due to the side-effects of existing antiallergic drugs and an elevated prevalence of allergic diseases, the challenges in discovering new allergy drugs continue, and natural products are suggested as one source of possible solutions [27,28].

*Callerya atropurpurea* (family: Fabaceae, genus: Adinobortys) belongs to the *Callerya* clade, and it is also known as *Adinobotrys atropurpures*. *C. atropurpurea* a tropical fruit tree species that inhabits Laos, Malaysia, Thailand, and Vietnam [29,30,31,32]. In the past, the local people used the twigs and roots of *C. atropurpurea* to stupefy fish and kill insects [29]. The anti-inflammatory effects of *C. atropurpurea* have been reported [31]. In this paper, it was found that Ca-EE can significantly suppress lipopolysaccharide-triggered nitric oxide production without exhibiting cytotoxicity [31]. In addition, this extract reduced TLR4 expression, changed MyD88/TRAF6 interaction, and altered the phospho-Src/PI3K/AKT and phosphor-MAPK pathways [31]. Under in vivo investigation, Ca-EE was revealed to reduce HCl/EtOH-induced gastric ulcer and LPS/poly (I:C)-induced septic shock [31]. We further investigated the antiallergic effects using the ethanol extract of *C. atropurpurea* (Ca-EE). This is because the allergic responses are mediated by inflammatory mediators and proinflammatory cytokines [4,33,34]. The molecular mechanism of Ca-EE inhibition of allergy was explored by quantitative PCR and Western blotting, and the in vivo efficacy of Ca-EE was examined in an atopic dermatitis model.

## 2. Results

### 2.1. GC/MS Analysis of Callerya atropurpurea (Ca-EE)

To investigate the individual components of the 70% ethanol extract of *Callerya atropurpurea* (Ca-EE), we performed GC/MS (Figure 1) analysis with 50 mg/mL Ca-EE in 100% DMSO. Through this approach, we identified 26 phytochemical compounds in Ca-EE, as listed in Table 1. Thus, it was revealed that Ca-EE contains 12.673% cis-raphasatin, 10.578% lupeol, 10.508% phytol, 9.755% 9,12,15-octadecatrienoic acid, ethyl ester, 8.812% ethyl tridecanoate, 7.592% xylitol, and 6.904% trehalose. In addition, sugars (d-mannose, trehalose) and fatty acids (hexadecanoic acid, heptadecanoic acid, linoelaidic acid, and octadecatrienoic acid) were also minorly detected. 

### 2.2. Antiallergic Activities of Ca-EE in RBL-2H3 Cells

We aimed to explore the role of Ca-EE in allergic responses. First, the inhibitory action of Ca-EE on lipoxygenase was examined. Lipoxygenase is the main enzyme that synthesizes inflammatory metabolites in atopic disease [35]. Ca-EE downregulated lipoxygenase activity up to 43% at 100 μg/mL, and the well-known lipoxygenase inhibitor, quercetin [36], was used as a control in the assay (Table 2). Next, the level of β-hexosaminidase release was tested. IgE-sensitized RBL-2H3 cells, adherent cells with histamine-releasing function [37,38], were incubated with Ca-EE, and HSA stimulated the degranulation of β-hexosaminidase. Ca-EE significantly inhibited β-hexosaminidase release from mast cells at 25–100 μg/mL (Figure 2a) without cell cytotoxicity (Figure 2b). We found that Ca-EE regulated allergic responses in RBL-2H3 mast cells.

### 2.3. Inhibitory Action of Ca-EE on Allergic Inflammatory Cytokine Transcription and the IgE-FceRI Signaling Pathway

To determine whether Ca-EE exhibited antiallergic activity at the transcriptional level, quantitative PCR was performed. Allergic response-related inflammatory cytokines including IL-4, TNF-α, IL-1β, and TGF-β1 were observed in IgE- and HSA-treated RBL-2H3 cells. Ca-EE showed transcription regulatory roles in these genes in a dose-dependent manner (Figure 3a). Next, the IgE/FcεRI-stimulated signaling pathway was screened. As a result, Ca-EE suppressed PKCδ activities (Figure 3b), and phosphorylated AKT was also decreased (Figure 3c). Upstream molecules of PKCδ and AKT, i.e., Lyn and Syk, were analyzed, which could interact with FcεRI. Ca-EE markedly downregulated the phosphorylation levels of Lyn and Syk at 100 μg/mL of this extract (Figure 3d). Overall, Ca-EE exhibited antiallergic effects by modulating IgE/FcεRI-mediated signaling pathways.

### 2.4. In Vivo Effects of Ca-EE in the DNCB-Induced Contact Dermatitis Model

To determine if in vitro efficacy would translate to an in vivo model, we conducted a DNCB-mediated atopic dermatitis mouse model study (Figure 4a). Ears of mice were pretreated with Ca-EE or dexamethasone for 7 days. From day 7, DNBC (0.5–1%) was topically applied every 3 days to the same spots to induce atopic dermatitis. As shown in Figure 4b, DNCB caused ear swelling, hyperemia, and hemorrhage. Application of Ca-EE distinctly recovered the ear appearance from the atopic reaction without body weight loss (Figure 4c). The dermatitis score was evaluated, and application of Ca-EE alleviated symptoms—itching or eruption—at a comparable rate to dexamethasone (Figure 4d). The DNCB-treated group had thickened ears, while thinner ears were observed in the Ca-EE-treated groups (Figure 4e). However, the dryness of DNCB-applied ear lesions was not altered between experimental groups (Figure 4f).

### 2.5. Histological and Immunological Analysis of Ca-EE-Mediated Antiatopic Effects in the DNCB-Induced Dermatitis Model

Collected ears and the serum of DNCB-induced murine dermatitis model mice were histologically and immunologically analyzed. The thickness and inflammation of atopic lesions were determined with H&E staining (Figure 5a). Ear skin thickness was increased in the DNCB-treated group compared with the normal group. Ca-EE treatment significantly restored ear thickness and inflammatory lesions to a normal level. Tissues were stained with toluidine blue to confirm mast cell infiltration into atopic lesions (Figure 5b). As shown, the DNCB group had significantly increased mucosal mast cells in the dermis, but administration of Ca-EE lowered the number of mast cells (Figure 5c). These results indicate that Ca-EE markedly suppressed mast cell activation and migration.

IgE and IL-4 are key mediators of allergic inflammation [39]; thus, the levels of IgE in serum and IL-4 in ear were evaluated. Increased IgE production and IL-4 gene expression in atopic dermatitis were significantly reduced after topical Ca-EE application (Figure 5d,e).

## 3. Discussion

In this study, we explored the antiallergic activities of Ca-EE in both in vitro assays and an in vivo model. GC/MS analysis showed that Ca-EE contained 12.673% cis-raphasatin, 10.578% lupeol, 10.508% phytol, 9.755% 9,12,15-octadecatrienoic acid, ethyl ester, 8.812% ethyl tridecanoate, 7.592% xylitol, and 6.904% trehalose (Figure 1 and Table 1). In in vitro assays, Ca-EE significantly reduced the activities of lipoxygenase and β-hexosaminidase (Figure 2a,b). The transcription of inflammatory genes (IL-4, TNF-α, IL-1β, and TGF-β1) and the activation of Lyn was suppressed by Ca-EE in IgE-induced RBL-2H3 cells (Figure 3 and Figure 4). Although we did not check protein levels of these cytokines, it is expected that secreted levels of these cytokines can be reduced. This is because Ca-EE can strongly suppress the upstream signaling event composed of Lyn and its downstream enzymes, involved in the regulation of transcriptional and translation responses of proinflammatory cytokines. In addition, the in vitro efficacy of Ca-EE was translated into an in vivo atopic dermatitis model, and histological analysis and molecular dissection were performed (Figure 5 and Figure 6). In line with the previously published literature, lupeol, phytol, and octadecatrienoic acid seem to be active ingredients in Ca-EE with antiallergic function. This is because lupeol was reported to display reduction in IL-4 and IL-5 in BALB/c mice immunized with ovalbumin [40]. Phytol is known to have anti-inflammatory, antioxidant, autophagy- and apoptosis-inducing, antinociceptive, immune-modulating, and antimicrobial effects [41]. Octadecatrienoic acid inhibits anaphylaxis by downregulating Lyn kinase phosphorylation [42]. To prove this, whether these molecules are really detected in Ca-EE and whether these compounds can indeed show antiallergic activity should be determined by HPLC and confirmed under in vitro experimental conditions.

After exposure of specific allergens, an inflammatory reaction is provoked—allergic inflammation—and Th2 cytokines are secreted [43,44]. Our recent report showed that the anti-inflammatory efficacy of Ca-EE targeted the Toll like receptor (TLR) 4 in its ligand lipopolysaccharide (LPS)-mediated signaling—MAPK/AP-1 and PI3K/AKT signaling [31]. TLRs can recognize environmental allergens and manifest an immune reaction. In particular, TLR4 activation altered the Th1/Th2 balance, resulting in amplification of allergic diseases [45]. *Saposhnikovia divaricata* aqueous extracts and daphnetin (a bioactive coumarin derivate) displayed anti-inflammatory and antiallergic effects in allergic rhinitis (AR) models through modulating TLR4/NF-κB signaling [46,47]. Moreover, regulation of FcεRI and TLR4 signaling potentiated DNCB-induced AD symptoms [48]. Regulation of TLR4-mediated cascades could be one mechanism of antiallergic effects. We could expect effective antiallergic roles of Ca-EE through anti-inflammatory functions.

We explored the IgE/FcεRI signaling pathway. Lyn and related kinases such as Syk, AKT, and PKCδ were significantly inhibited by Ca-EE treatment (Figure 4b–d). Lyn kinase belongs to the Src family of kinases which are considered as initiating molecules of mast cell activation signaling [5,49]. Blocking of Lyn kinases in mast cells has the potential to suppress allergic responses. The natural flavonoid quercetin alleviated allergic conjunctivitis in a mouse model by inhibiting Lyn-mediated signaling [50]. One Src kinase family inhibitor, dasatinib (Sprycel^®^), was examined for an anti-allergic response in mast cells and in the passive cutaneous anaphylaxis in vivo model. Dasatinib directly interacted with Lyn, and degranulation of mast cells was inhibited [51]. As shown, Ca-EE effectively downregulated Lyn kinase phosphorylation, which is concluded to be a factor in its antiallergic activities. Similarly, the phosphorylation of its known downstream kinases such as Syk, PKCδ, and AKT [52,53] was strongly reduced by Ca-EE (Figure 3b–d). Nonetheless, since we did not confirm the actual role of this extract in the activation of Lyn and Syk with their antagonists or their genes; currently, we are unable to specify real target of Ca-EE. Therefore, further study related to the identification of target molecule(s) will follow.

For allergy treatment, diverse approaches are conducted in the fashion of controllers and relievers. Corticosteroids such as prednisolone, mometasone, and dexamethasone effectively relieve allergic symptoms with anti-inflammatory properties [27,54]. However, prolonged uptake of corticosteroids causes adverse effects such as hypertension, osteoporosis, edema, or adrenal suppression [26,55]. To overcome these adverse events, various trials have been attempted, and new targets are being suggested—IL-4 or IL-4R-targeting drugs (e.g., dupilumab) or immunotherapy [56,57]. Natural products are another option for allergy treatment, and there is evidence demonstrating therapeutic effects [28,58]. Natural products have been investigated as antiallergic substances. Not only plant extracts but also single compounds from plants improved allergic reactions in in vitro and in vivo studies [28]. For instance, *Clinacanthus nutans* leaf aqueous extract pharmacologically regulated allergic responses in ovalbumin-challenged active systemic anaphylaxis models [59]. Flavonoids including liquiritigenin, baicalein, and quercetin reduced the expression of AD effector molecules, IL-4 or TNF-α [58].

Prevalence of allergy is consistently increasing, with the associated higher burden of costs. In addition, allergy affects quality of life because patients cannot avoid allergens completely [60]. Natural products are often used as oral drugs that provide dose convenience. With effectiveness, natural products are safe with less anticipated toxicity and side effects [61]. In an atopic dermatitis mouse model, 4 weeks of repeated administration of Ca-EE did not lead to body weight loss (Figure 5c); however, an in-depth toxicity study of Ca-EE is required. As the safety of Ca-EE is assessed, the potential of Ca-EE as an anti-allergic drug is increased.

As summarized in Figure 6, in RBL-2H3 cells, the treatment of Ca-EE significantly downregulated the activation of lipoxygenase and β-hexosaminidase and cytokine expression. Furthermore, IgE-mediated signaling pathway was downregulated with Ca-EE by blocking Lyn/Syk/PKCδ/AKT kinases. Moreover, Ca-EE effectively inhibited allergic symptoms in the DNCB-induced atopic dermatitis mouse model. These results indicate that Ca-EE could be effective drug for allergic disease treatment.

## 4. Materials and Methods

### 4.1. Materials

RBL-2H3 cells (rat basophilic leukemia) were purchased from the American Type Culture Collection (ATCC) (Rockville, MD, USA). The cell culture media, trypsin (0.25%), and antibiotics (penicillin and streptomycin) were obtained from Hyclone (Logan, UT, USA). Dimethyl sulfoxide (DMSO), lipoxygenase from glycine max (soybean), DNP-IgE, DNP-HSA, 4-nitrophenyl-N-acetyl-β-D-glucosaminide, 1-chloro-2,4-dinitrobenzene (DNCB), toluidine blue O, 3-(4,5-dimethylthiazol-2-yl)-2,5-diphenyltetrazolium bromide (MTT), sodium carboxymethylcellulose (CMC), and formaldehyde were purchased from Sigma-Aldrich (St. Louis, MO, USA). Phosphate-buffered saline (PBS) was purchased from Samchun Pure Chemical Co. (Gyeonggi-do, Korea). The cDNA synthesis kit was purchased from Thermo Fisher Scientific (Waltham, MA, USA). All primers for checking the mRNA of markers were synthesized by Macrogen (Seoul, Korea). The qPCRBIO SyGreen Blue Mix Lo-ROX was obtained from PCR Biosystems, Inc. (Wayne, PA, USA). All antibodies used in this research were from Cell Signaling Technology (Beverly, MA, USA), except for β-actin, which was from Santa Cruz Biotechnology, Inc. (Dallas, TX, USA).

### 4.2. Animal Experiments

BALB/c mice were bought from Orient Bio Inc. (Seongnam, Gyeonggi-do, Korea). The animal experiments were performed under the guidance of the Institutional Animal Care and Use Committee at Sungkyunkwan University (Suwon, Korea; approval ID: SKKUIACUC2021-04-19-2).

### 4.3. Cell Culture

RBL-2H3 cells were grown in DMEM supplemented with 10% fetal bovine serum (Gibco, Grand Island, UT, USA) and 1% penicillin and streptomycin. The cells were incubated in a humidified incubator with 5% CO_2_ at 37 °C.

### 4.4. Ca-EE Preparation

Ca-EE (Code number: NIBR 1062) was obtained from the National Institute of Biological Resources (NIBR) (Incheon, Korea). The aerial parts of C. atropurpurea were collected in Bolikamxai, Laos PDR (Specimen number: NIBRVP0000814474). The dried branches and leaves of C. atropurpurea were soaked in 70% ethanol and extracted by an ultrasonic extractor (Ultrasonic Cleaner UC-10, UC-20) for 3 h at 50 °C. To remove ethanol and the aqueous solvent from C. atropurpurea crude extract, we evaporated alcohol layer with SpeedVac (ThermoFisher Scientific, Waltham, MA, USA), and lyophilized water layer with lyophilizer (Freeze Dryers, ThermoFisher Scientific, Waltham, MA, USA) at −80 °C for 48 h. Further evaporation of the aqueous solvent was lyophilized at 80 °C for 48 h. The extract was kept in a freezer compartment at −20 °C until use. The powder of Ca-EE (88.7 mg) was dissolved in DMSO to make a stock solution at the concentration of 50 mg/mL for in vitro experiments. For in vivo experiments, Ca-EE (4 mg/mL or 8 mg/mL) was prepared by suspension with 0.5% CMC and agitated before treatment.

### 4.5. Gas Chromatography/Mass Spectrometry

GC/MS analysis of Ca-EE was performed with an Agilent 8890 GC instrument (Santa Clara, CA, USA) equipped with an Agilent J&W DB-624 Ultra Insert GC column (60 m in length × 250 μm in diameter × 1.40 μm in thickness); mass spectrometry was conducted with an Agilent 5977B MSD instrument (Santa Clara, CA, USA), equipped with a Series II triple-axis detector with a high-energy dynode and long-life electron multiplier, from the Cooperative Center for Research Facilities of Sungkyunkwan University (Gyeonggi-do, Korea). The spectrum of phytochemicals in the National Institute of Standards and Technology library was used to identify the unknown phytochemicals in Ca-EE, as reported previously [62,63].

### 4.6. Cell Viability Assay

RBL-2H3 cells were seeded in a 96-well plate at a density of 5 × 10^5^ cells/mL and cultured for 18 h in an incubator. The cells were then treated with Ca-EE in concentrations of 25 μg/mL, 50 μg/mL, and 100 μg/mL. After 24 h, the cell viability was determined by a conventional MTT assay [64,65]. Six replicates were set in this assay.

### 4.7. Inhibition of 15-Lipoxygenase Activity Assay

The 15-LOX inhibitory activity of Ca-EE was determined according to the method described by Yasin et al. with a slight modification [66]. In addition, 15-LOXs promote the reaction between linoleic acid and oxygen, producing 13-hydroperoxyoctadecadienoic acid that can increase the absorbance at 234 nm. To prepare the mixture, 200 μL of sample (Ca-EE or quercetin) and 400 μL of soybean lipoxygenase solution (167 U/mL) were reacted in 3.2 mL of 100 mM sodium phosphate buffer (pH 7.4). The mixture was reacted at 25 °C for 10 min. Quercetin was used for positive control and prepared by omitting the sample from the mixture and adding only DMSO. The reaction was started by putting 200 μL of 2.5 mM sodium linoleic acid. The absorbance was measured at 234 nm every 30 s for 3 min using a UV/Vis spectrophotometer (BioTek Instruments Inc., Winooski, VT, USA). Lipoxygenase inhibitory activity was calculated by following equation:Lipoxygenase inhibitory activity (%) = [(∆A_1_/∆t − ∆A_2_/∆t)]/(∆A_1_/∆t)] × 100,
where ∆A_1_/∆t is the enzymatic activity in the control group, and ∆A_2_∆t is that in the sample group.

### 4.8. β-Hexosaminidase Activity Assay

RBL-2H3 cells were spread into 96-well plates at a density of 5 × 10^5^ cells/mL and incubated for 18 h. The cells were sensitized by treating with 100 ng/mL of anti-DNP IgE overnight. The cells were then washed with Siraganian buffer [119 mM NaCl, 5 mM glucose, 25 mM PIPES, 0.1 mM CaCl, 40 mM NaOH, 0.4 mM MgCl_2_, and 0.1% BSA (pH 7.2)] three times and treated with Ca-EE at concentrations of 25, 50, and 100 μg/mL After 30 min, DNP-HSA (10 μg/mL) was added to stimulate the cells for 1 h. This stimulation was stopped on ice for 5 min, and 50 μL of supernatants were transferred to a new plate. Next, 100 μL of 1 mM 4-nitrophenyl-N-acetyl-β-D-glucosaminide in 0.1 M sodium citrate (pH 4.5) was added to the supernatants and incubated for 2 h. The reaction was stopped by adding 100 μL of carbonate buffer containing 0.1 M NaHCO_3_ and 0.1 M Na_2_CO_3_ (pH 10). The absorbance of these solutions was measured at 405 nm using a microplate reader. Six replicates were set in this assay.

### 4.9. RNA Extraction and Quantitative Real-Time PCR

RBL-2H3 cells were seeded at a density of 5 × 10^5^ cells/mL in a six-well plate and incubated for 18 h. The cells were sensitized and washed as before in the β-hexosaminidase activity assay and treated with Ca-EE at concentrations of 50 and 100 μg/mL. After 24 h, total RNA was extracted with TRIzol reagent according to the manufacturer’s instructions [64]. The complementary DNA was made with Thermo Fisher Scientific’s cDNA synthesis kit. Quantitative real-time PCR was performed with Pcrbio’s qPCRBIO SyGreen mix. All the primers are listed in Table 3.

### 4.10. Preparation of Cell Lysates and Immunoblotting Analysis

Total lysates of RBL-2H3 cells were prepared using lysis buffer (50 mM Tris-HCl (pH 7.5), 120 mM NaCl, 25 mM β-glycerol phosphate (pH 7.5), 20 mM NaF, 2% NP-40, 2 μg/mL aprotinin, 2 μg/mL leupeptin, 2 μg/mL pepstatin A, 1 mM benzamide, 1.6 mM pervanadate, 100 μM PMSF, and 100 μM Na_3_VO_4_). The protein concentration of cell lysates was measured using the Bradford assay, and loading samples were prepared for SDS-PAGE. The proteins were transferred from SDS-PAGE gel to a PVDF membrane. Immunoblotting analysis was used to target the phosphorylated and total forms of PKCδ, PLCγ, AKT, Lyn, Syk, and β-actin. Each antibody was incubated with the PVDF membrane in 3% BSA in TBST at a ratio of 1:2500 at 4 °C for 18 h. The second antibodies were then incubated for 2 h at a ratio of 1:2500 at RT. The immunoreactive bands were detected by EDP (enhanced peroxidase detection) of ELPIS-BIOTECH in a Chemidoc of ATTO.

### 4.11. DNCB-Induced Atopic Dermatitis Mouse Model

BALB/c mice (male, 4 weeks old) were acclimated for 1 week upon arrival. Mice were randomly divided into five groups of five mice (only PBS, only DNCB, DNCB, 4 mg/mL Ca-EE, DNCB and 8 mg/mL Ca-EE, and DNCB and 8 mg/mL dexamethasone). DNCB, Ca-EE (200 μL/ear), and dexamethasone (200 μL/ear) were sequentially dissolved in PBS and applied to both ears of mice. Ca-EE and dexamethasone were treated at 200 μL/ear once a day for 28 days. On day 7, 200 μL of 1% DNCB was applied to each ear, and treatment with 200 μL of 0.5% DNCB was repeated every 3 days. On Day 28, all the mice were euthanized, and blood samples and ears were collected. The serum was isolated from the blood samplem and an IgE-ELISA assay was performed with an ELISA kit (BD Biosciences, Oxford, UK) according to the manufacturer’s instructions.

### 4.12. Measurements of Water Content

The water content (%) of the ears of the mice was measured using an SK-IV digital moisture monitor. The measurements were performed twice a week, and the water content value of each mouse was made by averaging the two ears.

### 4.13. Histopathological Analysis

After isolation from each mouse, ears were fixed in 4% formaldehyde solution for 2 days. The ear samples were then embedded in paraffin, and the paraffin blocks were cut to 4 μm thick slides for H&E staining or toluidine blue O staining. Toluidine blue O staining started with deparaffinization of paraffin slides by xylene. The slides were then hydrated with distilled water. The slides were immersed into toluidine blue solution for 2–3 min, washed with distilled water three times, and then dehydrated with 95% and 100% alcohol. Lastly, the slides were dipped into xylene twice, for 3 min each. For counting mucosal mast cells, toluidine blue *O*-stained cells were counted with a cell counter.

### 4.14. Quantification of IgE Level in Mouse Serum

After the mice were sacrificed, the blood was extracted from the mice. The blood was centrifuged with 3000 RPM for 15 min and the supernatant was collected. For quantification of IgE level in mouse serum, the ELISA kit of Invitrogen (Waltham, MA, USA) was used.

### 4.15. Statistical Analysis

All data acquired from this research are presented as the mean ± standard deviation of at least three independent experiments. In animal experiments, the body weight, water content, dermatitis score, and ear thickness were analyzed by two-way ANOVA using GraphPad Prism 8 statistics software (Ver. 8, GraphPad Software, San Diego, CA, USA). All other data were analyzed with the Mann–Whitney test. Statistical significance was judged with *p*-value less than 0.05.

## Figures and Tables

**Figure 1 plants-12-00860-f001:**
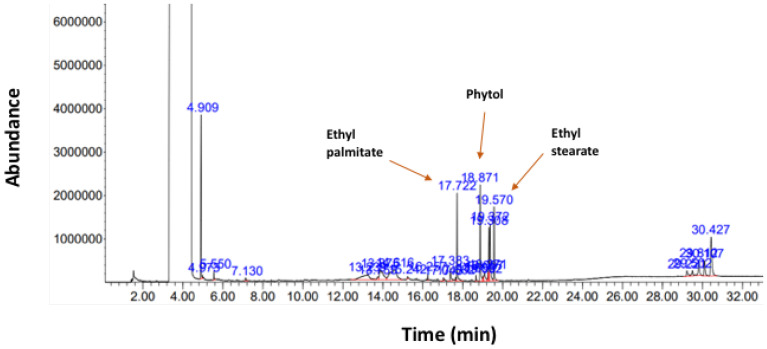
Phytochemical profiling of *Callerya atropurpurea* ethanolic extract (Ca-EE). Identified compounds are listed in Table 1.

**Figure 2 plants-12-00860-f002:**
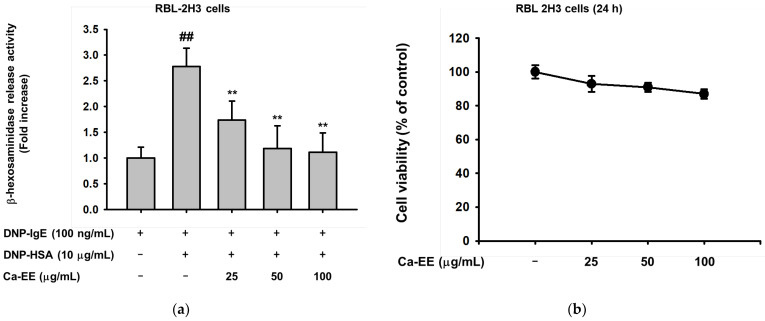
The effect of Ca-EE on β-hexosaminidase release in IgE- and HSA-treated RBL-2H3 cells. (**a**) RBL-2H3 cells were incubated with DNP-IgE and DNP-HSA to induce β-hexosaminidase activity. Cells were treated with Ca-EE at the indicated concentration. The supernatant of the cell culture was transferred to a plate and reacted with glucosaminide for 2 h. β-Hexosaminidase release was examined by measuring absorbance at 405 nm. (**b**) Cell viability of RBL-2H3 cells under Ca-EE treatment (0–100 μg/mL). Six replicates were set in these assays (**a**,**b**). The conventional MTT assay was performed **^##^**
*p* < 0.01 compared with the normal group; ** *p* < 0.01 compared with the control group.

**Figure 3 plants-12-00860-f003:**
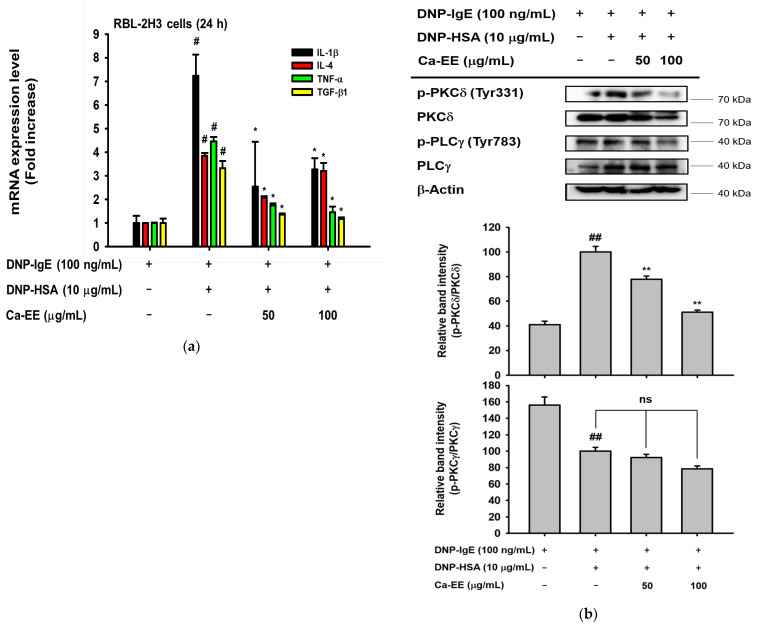
Regulation by Ca-EE of inflammatory mediators in IgE- and has-treated RBL-2H3 cells. (**a**) Ca-EE was incubated with IgE ahasHSA for 24 h in RBL-2H3 cells. The mRNA levels were isolated from cells, and quantitative PCR was conducted. Primers for IL-1β, IL-4, TNF-α, and TGF-β1 were used. Six replicates were set in this analysis. (**b**–**d**) The effects of Ca-EE on allergic inflammatory signaling. RBL-2H3 cells were pre-incubated with Ca-EE for 30 min, and then treated with DNP-IgE (100 μg/mL) and DNP-HAS (10 μg/mL) to induce an allergic response. After 1 h, cells were harvested for immunoblotting analysis. Phosphorylated or total antibodies against PKCδ, PLCγ, AKT, Lyn, Syk and β-actin were used. β-actin was used as a loading control. The band intensity (middle and lower panels of b, and lower panels of c and d) of p-PKCδ, p-PLCγ, p-AKT, p-Lyn, and p-Syk was measured with their total forms using the Image J program, and the three replicates were used. **^#^**
*p* < 0.05 and **^##^**
*p* < 0.01 compared with the normal group; * *p* < 0.05 and ** *p* < 0.01 compared with the control group. ns: not significant.

**Figure 4 plants-12-00860-f004:**
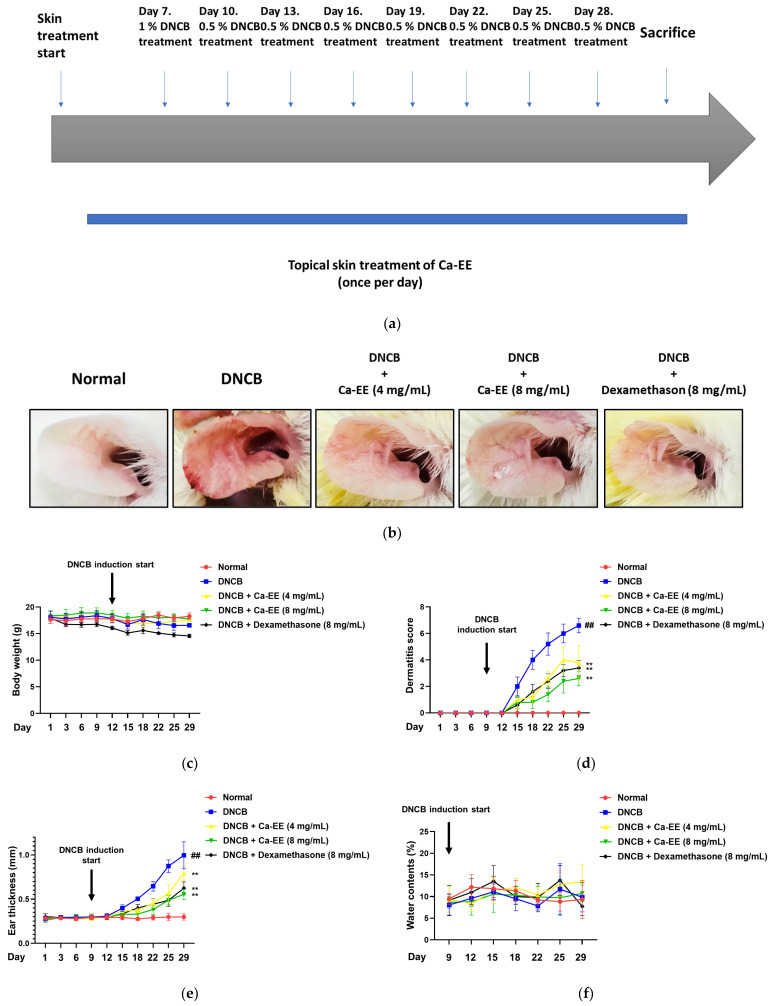
In vivo efficacy of Ca-EE in an atopic dermatitis (AD) mouse model. (**a**) Schematic diagram of the AD mouse disease model. Ca-EE was topically applied to skin once a day for 7 days before DNCB treatment. AD was induced by DNCB treatment for an additional 21 days. (**b**) Ears of AD-induced mice. Photographs were taken at day 28. (**c**) The dermatitis score was determined by changes of itching, erythema, and dryness of lesion (Criteria 0: no change, 2: slight, 4: moderate, 8: severe change). (**d**) The ear thickness of mice was measured by dial thickness gauges. (**e**) The body weight of mice was measured every 3 days. (**f**) The water content (%) of skin was evaluated using an SK-IV digital moisture monitor. **^##^**
*p* < 0.01 compared with the normal group; ** *p* < 0.01 compared with the control group.

**Figure 5 plants-12-00860-f005:**
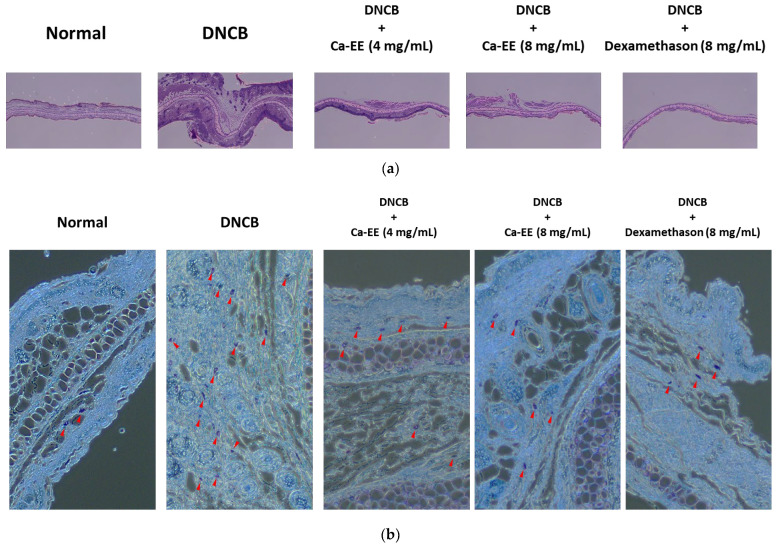
In vivo effect of Ca-EE in the DNBC-induced AD mouse model. (**a**,**b**) In DNBC-induced AD disease model mice, ears and blood were collected for hematological analysis and stained with hematoxylin and eosin (H&E) (**a**) and toluidine blue O (**b**). (**c**) The numbers of toluidine blue O-stained mast cells were counted with a cell counter. (**d**) Serum was isolated, and the levels of IgE were examined by ELISA. (**e)** The mRNA level of IL-4 in ear lysates was detected by real-time quantitative PCR. **^##^**
*p* < 0.01 compared with the normal group; ** *p* < 0.01 compared with the control group.

**Figure 6 plants-12-00860-f006:**
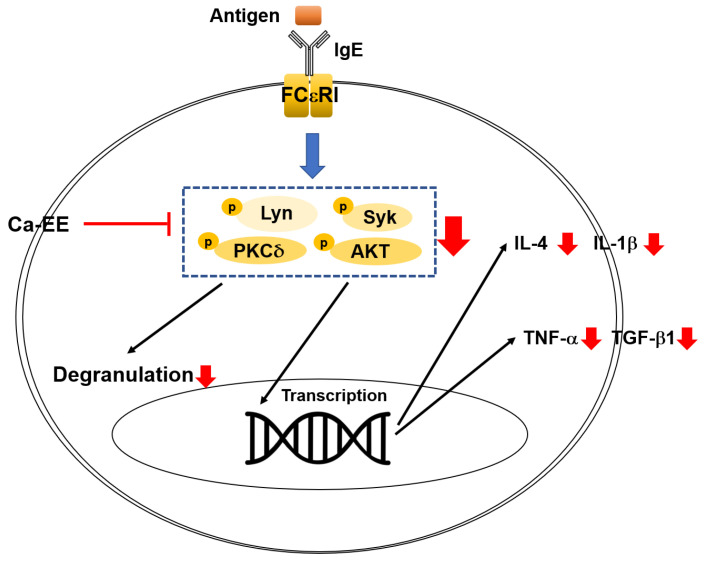
Schematic summary of the antiallergy effects of Ca-EE and its mechanisms.

**Table 1 plants-12-00860-t001:** The identified compounds in Ca-EE.

Peak No.	Name of Compound	% of Total
1	Methanesulfonyl fluoride	0.275
2	Succinic acid, di(3-phenylprop-2-en-1-yl) ester	0.752
3	Oxalic acid, 4-chlorophenyl octadecyl ester	0.569
4	Xylitol	7.592
5	d-Mannose	0.802
6	Trehalose	6.904
7	cis-Raphasatin	12.673
8	1-Hydroxy-1-(4-methoxyphenyl)propan-2-one	0.396
9	2-Pentadecanone, 6,10,14-trimethyl	0.625
10	Octahydropyrrolo[1,2-a]pyrazine	0.351
11	n-Hexadecanoic acid	2.047
12	Ethyl 9-hexadecenoate	0.356
13	Ethyl tridecanoate	8.812
14	Heptadecanoic acid, ethyl ester	0.536
15	Phytol	10.508
16	Linoelaidic acid	0.635
17	9,12,15-Octadecatrienoic acid, (Z,Z,Z)-	2.334
18	Octadecanoic acid	0.953
19	Linoleic acid ethyl ester	4.979
20	9,12,15-Octadecatrienoic acid, ethyl ester, (Z,Z,Z)-	9.755
21	Octadecanoic acid, ethyl ester	7.126
22	Brallobarbital	1.655
23	Cyclohexane-1,3-dione, 2-allylaminomethylene-5,5-dimethyl-	1.437
24	beta.-Amyrin	3.852
25	Olean-12-ene	3.501
26	Lupeol	10.578

**Table 2 plants-12-00860-t002:** The effect of Ca-EE on lipoxygenase activity.

Samples (μg/mL)		Inhibition of Lipoxygenase (%)
Ca-EE	50	40.1 ± 1.5
	100	43.9 ± 3.8
Quercetin	10	34.8 ± 3.0

Quercetin was used as a positive control. Three replicates were set in this assay.

**Table 3 plants-12-00860-t003:** List of quantitative PCR primers used for mRNA analysis.

Gene	Species	Direction	Sequence
*IL-4*	Rat	F	GTACCAGACGTCCTTACGGC
		R	ATTCACGGTGCAGCTTCTCA
*TNF-a*	Rat	F	GGCTTTCGGAACTCACTGGA
		R	GGGAACAGTCTGGGAAGCTC
*IL-1β*	Rat	F	TTGAGTCTGCACAGTTCCCC
		R	TCCTGGGGAAGGCATTAGGA
*TGF-β1*	Rat	F	AGGGCTACCATGCCAACTTC
		R	CCACGTAGTAGACGATGGGC
*IL-4*	Mouse	F	ATGGATGTGCCAAACGTCCT
		R	AAGCCCGAAAGAGTCTCTGC

## Data Availability

The data used to support the findings of this study are available from the corresponding author upon request.

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
