# Peer review of "Antiallergic Effects of Callerya atropurpurea Extract In Vitro and in an In Vivo Atopic Dermatitis Model"

_plants, 2023, doi:10.3390/plants12040860_

Round 1

Reviewer 1 Report

Wooram Choi and colleagues reported Callerya atropurpurea extract showed strong anti-inflammatory activity.

The data is solid.

I recommend to improve Abstract. For example, "but the roles in allergic responses" → Although ...., the molecular mechanisms of .....

Please describe on Callerya atropurpurea more in Introduction.

Author Response

Reviewer #1

Wooram Choi and colleagues reported Callerya atropurpurea extract showed strong anti-inflammatory activity.

The data is solid.

I recommend to improve Abstract. For example, "but the roles in allergic responses" → Although ...., the molecular mechanisms of .....
***Answer: We agreed with you. We tried to improve abstract (see L15-28).

Please describe on Callerya atropurpurea more in Introduction.
***Answer: Thank you for the comment. We added more information of C. atropurpurea in introduction part (Line 69-75).

Reviewer 2 Report

The article presents the investigation of anti-allergic effects of Callerya atropurpurea ethanolic extract. The extract is characterized by GC-MS (qualitative, not quantitative) and studies are made on its effect in vitro and in vivo.

I believe the article would benefit if the following information would be added:

Exact composition of the extract.

What particular compounds present anti-allergic effects?

How are the other compounds affect its action? 

Would purification be necessary for administration? 

Histamine Release Assay

Author Response

Reviewer #2

The article presents the investigation of anti-allergic effects of Callerya atropurpurea ethanolic extract. The extract is characterized by GC-MS (qualitative, not quantitative) and studies are made on its effect in vitro and in vivo.

I believe the article would benefit if the following information would be added:

Exact composition of the extract.
***Answer: The ethanol extract of Callerya atropurpurea was analyzed by GC-MS, and result are shown in Table 1. All peaks except DMSO (a peak before first one) are defined, and percent of each compound in Ca-EE is listed up (see Fig. 1 and Table 1).

What particular compounds present anti-allergic effects?
How are the other compounds affect its action? 
***Answer: Unfortunately, we cannot test the anti-allergic action of all identified compounds, but we found that some compounds such as lupeol, phytol, and octadecatrienoic acid were identified as anti-allergic components. Relevant sentences have been included in L200-208.

Would purification be necessary for administration? 
***Answer: We appreciate you for this comment. For preparing in vivo Ca-EE solution, we weighted 40 or 80 mg from the powder of dried Ca-EE (the ethanolic extract of C. atropurpurea leaves and stem) and mixed with 0.5% CMC to prepare 4 mg/mL or 8 mg/mL solution and continuously agitated until these are treated. 200 mL of Ca-EE solution (4 mg/mL or 8 mg/mL) was applied to each ear. Therefore, purification step is not necessary. Relevant sentences have been included in L302-305 and 367.

Histamine Release Assay
***Answer: Thank you for this comment. In this study, we did not detect the histamine level, but we performed b-hexosaminidase and lipoxygenase assays at the cellular and enzyme levels. These are well-known experimental conditions in testing anti-allergic activity of a compound or an extract. And also, these two enzymes are reported to be closely connected in histamine release process. Therefore, inhibition of b-hexosaminidase and lipoxygenase is accepted as another concern as the reduction of histamine release. Although we did not directly check histamine level, inhibitory effects on these two enzymes strongly indicate its suppressive activity of histamine and leukotriene releases. We really apologize for this, but if you still need this, then we will do it again.

Reviewer 3 Report

This paper describes a valuable work, performed taking account both in vitro and in vivo models.

Several comments can be provided, highlighting strength and limits.

Introduction:

if the introduction on allergic response and the link between chronic allergy and atopic dermatitis is clearly outlined, the part on the studied species is weak and need to be enlarged, reporting the state of art of the knowledge: how Callerya atropurpurea is used in traditional medicine? what's the regolatory status in Asia and in other countries? How the species is used and for what? What does the drug contain? What about the published literature?

Materials and methods:

Chemical analyses are not reported and this is a big, big pifall: also in results the chemical characterization is very poor; for sure an ethanol extract mainly contains phenolics, and/or di- or triterpenes, and/or alkaloids and, as your submission is intended for the section "phytochemistry", an accurate HPLC-DAD analysis is mandatory. Also GC analysis is poor, as we can't understand if reference standards or Kovats index have been used to identify and quanitfy different compounds.

Other results:

The considered markers are interesting and, despite a simple in vitro model, results obtained are consistent.

To be honest, a protein dosage to detect the downregulation of released cytokines would improved the quality of the work (mRNA level should always be confirmed because signal acrivation and effector release are two different things in biology); to be realistic, on the other hand, I ask authors to discuss this gap and add this limitation in results.

Again in results: insert KDa for each protein in WB analysis.

A question: did authors make a time course to detect the peak of activated p-proteins? The 24h is very long and not the best, I suppose, in an in vitro model.

The in vivo model is validated and results are clear in my opinion, but I admit I'm not as trained as in in vitro molecular pharmacology.

Discussion:

It is good. It may be finely improved by discussing the progress provided by this work compared to previous literature on the species.

But... I suggest to modify the graphical summary: as you did not use a selective antagonist and confirm the actual role of the extract in Lyn->Syk blockage, you can just hypotesize this mechanism.  

Author Response

Reviewer #3

This paper describes a valuable work, performed taking account both in vitro and in vivo models. Several comments can be provided, highlighting strength and limits.

Introduction:

if the introduction on allergic response and the link between chronic allergy and atopic dermatitis is clearly outlined,

***Answer: Thanks very much for the comment. According to this comment, we have added additional sentences in L48-51.

the part on the studied species is weak and need to be enlarged, reporting the state of art of the knowledge: how Callerya atropurpurea is used in traditional medicine? what's the regulatory status in Asia and in other countries? How the species is used and for what? What does the drug contain? What about the published literature?
***Answer: We appreciate you for the comments. We have tried to expand the information about Callerya atropurpurea in Introduction part (see L69-75). The tree can be found throughout East Asia, and its fruits and leaves are edible. One publication was only found, which is published by our group, anti-inflammatory roles of Ca-EE in LPS-induced inflammation models. Our aim of this study was to explore a new immunopharmacological usage of Callerya atropurpurea related to inflammation and immunomodulation, since it was reported to reduce the production of pro-inflammatory cytokines and inflammatory mediators.

Materials and methods:

Chemical analyses are not reported and this is a big, big pifall: also in results the chemical characterization is very poor; for sure an ethanol extract mainly contains phenolics, and/or di- or triterpenes, and/or alkaloids and, as your submission is intended for the section "phytochemistry", an accurate HPLC-DAD analysis is mandatory. Also GC analysis is poor, as we can't understand if reference standards or Kovats index have been used to identify and quanitfy different compounds.
***Answer: Thanks very much for your valuable comment. We totally agree with your comment. First, we hoped to have phytochemical fingerprinting profile of Ca-EE for its industrial application, since it showed promising effect on allergic reaction. Since GC-Mass analysis can be easily performed in our University and we believed that we can easily get chemical information on some of peaks detected. To analyze this, we have prepared Ca-EE solution (50 mg/mL in 100% DMSO) and ionized it to analyze (a 70eV electron is used for ionization). Although we could not have many peaks with higher contents, it is thoughtful that currently observed peaks can be applied for phytochemical fingerprinting compound(s) and some of these can be considered as active components based on several papers. Due to time limitations, unfortunately, we are not able to provide HPLC-DAD analysis. Nonetheless, lupeol and phytol seem to be considered as a standard or marker chemical for the standardization of this extract.

Moreover, as you mentioned, knowing which compounds can be working as active components should be further confirmed. For this, simply buying purified or standard commercial compounds among identified ones or activity-guided fractionation and purification of active compounds should be followed. But, as you easily know, these also require for more time to get or prepare. Therefore, we would like to ask you whether you can accept current approaches with GC-Mass analysis and whether you can wait until we have more experiments related to get these compounds at the second report. Regardless, we really apologize for this.

Relevant sentences to support this notion have been included in L83-85, 188-190, and L200-208.

Other results:

The considered markers are interesting and, despite a simple in vitro model, results obtained are consistent. To be honest, a protein dosage to detect the downregulation of released cytokines would improved the quality of the work (mRNA level should always be confirmed because signal acrivation and effector release are two different things in biology); to be realistic, on the other hand, I ask authors to discuss this gap and add this limitation in results.
***Answer: Thanks very much for your comment. We totally agree with your comment. In this study, we did not detect the released cytokines using ELISA or other method at the protein levels, but we confirmed mRNA expression level in RBL-2H3 cells and IgE production in vivo model. Moreover, interestingly, upstream signaling event composed of Lyn, Syk, PKCd and AKT was also clearly suppressed by Ca-EE exposure. The mRNA level of cytokines was significantly decreased by Ca-EE treatment. Finally, in vitro efficacy was effectively translated into in vivo atopic model – alleviated dermatitis symptoms, ear thickness reduction and reduced IgE production, according to our data. Following the data, it is expected that Ca-EE can regulate the expression pathway of the pro-inflammatory cytokines as well. Therefore, inhibition of upstream signaling pathway by Ca-EE can let us easily speculate that translation of these genes and their protein levels will be reduced. Of course, we also agree that we have to check protein level of these cytokines using ELISA. However, time-limit and less budget to buy ELISA kits make this work unavailable. We really apologize this. Relevant sentences have been included in Discussion section (see L194-198).

Again in results: insert KDa for each protein in WB analysis.
***Answer: Yes, we have inserted KDa in each Western blotting result (see Fig. 3b, c, and d).

A question: did authors make a time course to detect the peak of activated p-proteins? The 24h is very long and not the best, I suppose, in an in vitro model.
***Answer: Thank you for your comment. This very good point. Although we agree with this comment, however, we just followed a condition previous reported [Lee HP, Choi W, Kwon KW, You L, Rahmawati L, Luong VD, Kim W, Lee BH, Lee S, Kim JH, Cho JY. Inhibitory Effects of Grewia tomentosa Juss. on IgE-Mediated Allergic Reaction and DNCB-Induced Atopic Dermatitis. Plants (Basel). 2022 Sep 27;11(19):2540], in which we incubated it for 24 h. Since we are currently trying to accumulate anti-allergic activity data of Cambodia-originated medicine plants, following previous condition seems to be important to compare their anti-allergic activities. Nonetheless, because we are totally accepting this comment, we will try to have data set observed in a time-dependent manner. We apologize for this.

The in vivo model is validated and results are clear in my opinion, but I admit I'm not as trained as in in vitro molecular pharmacology.

***Answer: Thanks very much for your comments on in vivo models. 

Discussion:

It is good. It may be finely improved by discussing the progress provided by this work compared to previous literature on the species.

***Answer: Thank you for the comment. Unfortunately, there were no published papers regarding anti-allergic activity of the same Genus or Species plants. Therefore, we are unable to discuss these points. We apologize this.

But... I suggest to modify the graphical summary: as you did not use a selective antagonist and confirm the actual role of the extract in Lyn->Syk blockage, you can just hypotesize this mechanism. 
***Answer: Thank you for the comment. Lyn/Syk activation is main signaling when IgE/FceRI is activated. Activated Lyn is known to directly phosphorylate Syk kinase. Though we did not confirm the direct inhibitory action of Ca-EE between Lyn and Syk, phospho-Syk seems to be reduced by blockade of Lyn kinase by Ca-EE. However, since we did not directly characterize this pathway with individual antagonists and genes under Ca-EE-treated conditions, we just modified Ca-EE to roughly target to these molecules. Therefore, we have modified Fig. 6 (see Fig. 6). Relevant mention has been included in L231-236.

Reviewer 4 Report

In the study, the authors studied the anti-allergic effects of the alcoholic extract of plant

 Callerya atropurpurea on a mast cell line  RBL-2H3 cells  and in vivo on a model of chemically induced dermatitis in mice.

The results showed promising effects on several markers of allergic-type inflammation and signalling pathways and contributed to the expansion of knowledge about the effects of the extract from this plant. However, the work has several shortcomings that must be removed before acceptance.

Introduction:

-          Characterize the mentioned plant: Latin name, family, genus and what type of plant it is. Add information why this plant was selected for study of antiallergic effect. Is the name Callerya atropurpurea – the scientific one?

-          Justify the suitability of cell line RBL-2H3 cells (rat basophilic leukemia) for study of mast cells allergic responses. Are these cells adherent?

Material and methods:

Line 309: It seems to me that 4 weeks old mice are too young for immunological experiments. Please comment on this.

Line 270: Correct this sentences“ Further evaporation of the aqueous solvent was lyophilize at 80 °C for 48 h.“ as it is not clear when ethanol was evaporated.

Line 271. what was the lyophilized extract dissolved in before adding to the cells? Were all the ingredients soluble?

Line 273: Add how many replicate wells were set in in vitro assays for each type of „treatment“.

Line 281: Explain what is the Siraganian buffer. I did not find the procedure how The effect of Ca-EE on lipoxygenase activity was determined in vitro (table 2 – data). Add this part and how % inhibition was calculated (from untreated control cells?

Line 302: there is no information on how the cell lysates were applied to the membranes, I assume that through classical SDS-PAGE analysis with subsequent application to the membrane. Complete and add the given information and whether it was a native or denatured gel. There is missing information how density of protein bands after immunoblotting procedure were quantified and what type of instrument was used. How many replicates were measured. Add this information to Legends to figures.

Line 324. I assume that after clarification in xylene, the sections were embedded in the mounting medium. Complete the information.

Fig 4f shows refers to „Water content (%) of skin was evaluated using an SK-IV digital moisture monitor. „ This method is not mentioned in this part. Please add.

RESULTS:

Line 84: name of plant Callerya atropurpurea should be written in italic.

Line 116: in text i tis mentioned „Next, the IgE-FceRI stimulated signalling pathway was screened.“ Then on Line 120 it is written „Overall, Ca-EE exhibited anti-allergic effects by modulating IgE/FceRI-mediated signalling pathways.“ Correct typographical error.

-Line 120: correct syntax of sentence“ was also affected with under 100 mg/mL Ca-EE treatment (Fig. 3d). Overall, Ca-EE exhibited..“.to following:„ was also affected with Ca-EE treatent under concentrations of 100 mg/mL“.

Fig.3: Why Relative band intensity was not measured for all immunoblotted proteins? There is missing figures for : p-PLC and p-Lyn, which were normalized to unphosphorylated form of proteins. Add these figures. Have you normalized also to beta actin? Why not?

Line 135: correct syntax of this sentence “ Ca-EE or dexamethasone  was pre-treated to mice ears for 7 d. „ for example“ Mice´s ears were pre-treated with Ca-EE or dexamethasone  for 7 days“.

Fig. 5b: What type of mast cells are stained with toluidine blue? Connective tissue mast cells or mucosal mast cells? Add this information to Methods.

Line 162-163: „As shown, the DNCB group had significantly increased mast cells in the dermis, but administration of Ca-EE lowered the number of mast cells.“ There is missing quantitative data (Figure) about mast cells counts on ears sections therefore how authors could state that they were significantly increased? No p value is shown. Add data from morphometric study.

Line 167: it is written“ IgE and IL-4 are key mediators of allergic inflammation [29], thus the levels of IgE and IL-4 in serum were evaluated. Increased IgE production and IL-4 gene expression in atopic dermatitis...“

However, in Fig. 5d data about IL-4 gene expression in blood cells is shown.  This is discrepancy.

It is not clear how mRNA expression of IL-4 in blood was performed.  What type of Kit was used and why IL-4 in serum was not measured by Elisa kit in parallel with determination of IgE antibodies by ELISA Kit? If possible add this data.  

Author Response

Reviewer #4

In the study, the authors studied the anti-allergic effects of the alcoholic extract of plant Callerya atropurpurea on a mast cell line RBL-2H3 cells and in vivo on a model of chemically induced dermatitis in mice. The results showed promising effects on several markers of allergic-type inflammation and signalling pathways and contributed to the expansion of knowledge about the effects of the extract from this plant. However, the work has several shortcomings that must be removed before acceptance.

***Answer: Thanks very much for your comments. We have revised according to your comments as below.

Introduction:

 -  Characterize the mentioned plant: Latin name, family, genus and what type of plant it is. Add information why this plant was selected for study of antiallergic effect. Is the name Callerya atropurpurea – the scientific one?
***Answer: We appreciated your opinion. C. atropurpurea is also known as Adinobotrys atropurpurea which belong to Fabaceae family and Adinobortrys genus. We have added this plant information in context (Line 69-71). In addition, why we chose this plant is because this plant was reported to have strong anti-inflammatory effect by our group (Ref: You L, Huang L, Jang J, Hong YH, Kim HG, Chen H, Shin CY, Yoon JH, Manilack P, Sounyvong B, Lee WS, Jeon MJ, Lee S, Lee BH, Cho JY. Callerya atropurpurea suppresses inflammation in vitro and ameliorates gastric injury as well as septic shock in vivo via TLR4/MyD88-dependent cascade. Phytomedicine. 2022 Oct;105:154338). Therefore, we employed this to test its anti-allergic activity and fortunately, we could get very good effect as you can see in Figures. As you mentioned, we have added the reason for choosing this plant. Please check the line 73-74.

-  Justify the suitability of cell line RBL-2H3 cells (rat basophilic leukemia) for study of mast cells allergic responses. Are these cells adherent?
***Answer: RBL-2H3 cells are common cell line to study allergic reaction as well as inflammation, because RBL-2H3 cells can release histamine reacted by IgE-FceRI interaction. RBL-2H3 cells are adherent cells. We shortly described RBL-2H3 cells in line 100-101.

Material and methods:

Line 309: It seems to me that 4 weeks old mice are too young for immunological experiments. Please comment on this.
***Answer: We are sorry for error. 4-week old mice were arrived and acclimated for 1 week. In in vivo study, 5-week old mice were used. Since we treat inducers and testing drugs for 4 weeks, we chose little younger mice for this experiment. Mistakes have been corrected (please see line 364-365).

Line 270: Correct this sentences“ Further evaporation of the aqueous solvent was lyophilize at 80 °C for 48 h.“ as it is not clear when ethanol was evaporated.
***Answer: Since C. atropurpurea were extracted in 70% ethanol, plant crude extract was evaporated to remove ethanol and remained water was removed by lyophilization. We have revised the sentence to make it clear (see L297-301).

Line 271. what was the lyophilized extract dissolved in before adding to the cells? Were all the ingredients soluble?
***Answer: Thanks for your comment. Dried extract was clearly dissolved in DMSO. Ca-EE in DMSO was used for cell-based assay. We also suspended dried extract in 0.5% CMC solution to prepare in vivo testing samples and this solution was agitated before treating it to ears. We have added this information in section 4.4 line 302-305.

Line 273: Add how many replicate wells were set in in vitro assays for each type of „treatment“.
***Answer: Six replicate well were set in b-hexosaminidase and cell viability assays and for each type of treatment. The other enzyme assay was performed with three replicates. Relevant sentences have been included in L107, 114, 134, 139, 310, and 340.

Line 281: Explain what is the Siraganian buffer. I did not find the procedure how The effect of Ca-EE on lipoxygenase activity was determined in vitro (table 2 – data). Add this part and how % inhibition was calculated (from untreated control cells?
***Answer: Thank you for your opinon. Siraganian buffer is composed of 119 mM NaCl, 5 mM glucose, 25 mM PIPES, 0.1 mM CaCl, 40 mM NaOH, 0.4 mM MgCl2, 0.1 % BSA, and pH is 7.2, as mentioned in L332-333. We wrote the composition of buffer in section 4.7 (see L303-304). As you mentioned, we have added lipoxygenase activity assay (as section 4.6) and calculation equation, as well (see L311-327).

Line 302: there is no information on how the cell lysates were applied to the membranes, I assume that through classical SDS-PAGE analysis with subsequent application to the membrane. Complete and add the given information and whether it was a native or denatured gel. There is missing information how density of protein bands after immunoblotting procedure were quantified and what type of instrument was used. How many replicates were measured. Add this information to Legends to figures.
***Answer: Thank you for your comment. The detail information of immunoblotting analysis was added in section 4.9 (see L355-357). We measured band density using Image J program, and three triplicates were used for the analysis. This information is added in figure legend (Figure 3: L138-140)

Line 324. I assume that after clarification in xylene, the sections were embedded in the mounting medium. Complete the information.
***Answer: Yes, we agreed with you. Additional explanation has been added (see L382-383).

Fig 4f shows refers to „Water content (%) of skin was evaluated using an SK-IV digital moisture monitor. „ This method is not mentioned in this part. Please add.
***Answer: As you mentioned, “Measurement of water content” has been added in section 4.11 (see L374-377).  

RESULTS:

Line 84: name of plant Callerya atropurpurea should be written in italic.
***Answer: We apporeciate for your comment. We have revised it as well as others (see L30 and 80).

Line 116: in text i tis mentioned „Next, the IgE-FceRI stimulated signalling pathway was screened.“ Then on Line 120 it is written „Overall, Ca-EE exhibited anti-allergic effects by modulating IgE/FceRI-mediated signalling pathways.“ Correct typographical error.
***Answer: Yes, we have missed to type in greek alphabet. We corrected it (see L124).

Line 120: correct syntax of sentence“ was also affected with under 100 mg/mL Ca-EE treatment (Fig. 3d). Overall, Ca-EE exhibited..“.to following:„ was also affected with Ca-EE treatent under concentrations of 100 mg/mL“.
***Answer: Thank you for kind revision. We revised the sentence (Please see line 127-128).

Fig.3: Why Relative band intensity was not measured for all immunoblotted proteins? There is missing figures for : p-PLC and p-Lyn, which were normalized to unphosphorylated form of proteins. Add these figures. Have you normalized also to beta actin? Why not?
***Answer: Thank you for comment. We did not normalize to beta actin because we think that normalizing to total form than normalizing to beta actin represents relative level of phosphorylation more precisely. We also added the band density of p-Syk and p-PLCg. Relevant figures and sentences have been included in bottom panels of Fig. 3b and Fig. 3d, and L138-140.

Line 135: correct syntax of this sentence “ Ca-EE or dexamethasone  was pre-treated to mice ears for 7 d. „ for example“ Mice´s ears were pre-treated with Ca-EE or dexamethasone  for 7 days“.
***Answer: As you mentioned, we have revised the sentence (line 145-146).

Fig. 5b: What type of mast cells are stained with toluidine blue? Connective tissue mast cells or mucosal mast cells? Add this information to Methods.
***Answer: In Fig. 5b, we stained the mucosal mast cells. We indicated which mast cells are stained in samples in manuscript (L173 and 385-387).

Line 162-163: „As shown, the DNCB group had significantly increased mast cells in the dermis, but administration of Ca-EE lowered the number of mast cells.“ There is missing quantitative data (Figure) about mast cells counts on ears sections therefore how authors could state that they were significantly increased? No p value is shown. Add data from morphometric study.
***Answer: We agreed with you. The number of mast cells in staining results was counted and graphed in Fig. 5c. According to the graph, the mast cell number was significantly decreased by Ca-EE with p value < 0.05. Relevant sentences and data have been included in Fig. 5c, L174, and L385-387.

Line 167: it is written“ IgE and IL-4 are key mediators of allergic inflammation [29], thus the levels of IgE and IL-4 in serum were evaluated. Increased IgE production and IL-4 gene expression in atopic dermatitis...“
However, in Fig. 5d data about IL-4 gene expression in blood cells is shown. This is discrepancy.
It is not clear how mRNA expression of IL-4 in blood was performed. What type of Kit was used and why IL-4 in serum was not measured by Elisa kit in parallel with determination of IgE antibodies by ELISA Kit? If possible add this data.
***Answer: We are sorry to make you confused. We examined IL-4 expression level using ears but not serum. So that, measurement of mRNA level was better than ELISA. We corrected it in figure legend and manuscript (see Fig. 5d, and L176-177, and L183). As you mentioned, method of IgE production measurement was added as section 4.13 in materials and method part (see L388-392).

Round 2

Reviewer 3 Report

I truly appreciate the honest comments of authors as reply to reviewers and I believe that they are trying to do their best.

With the aim to be propositive, I need to comment that authors miss to include GC analysis in M&M, please add this paragraph.

If possible, a very minor point, enlarge introduction describing the regulatory status of the species in your country and some information retrieved by tradition and ethnobotany.

Thank you.  

Author Response

Reviewer #3.

I truly appreciate the honest comments of authors as reply to reviewers and I believe that they are trying to do their best.

With the aim to be propositive, I need to comment that authors miss to include GC analysis in M&M, please add this paragraph.

***Answer: Thank you for the comment. Sorry for that we have missed to include the GC analysis in M&M. We added the paragraph about GC analysis in 4. Materials & Methods section (see L313-322).

If possible, a very minor point, enlarge introduction describing the regulatory status of the species in your country and some information retrieved by tradition and ethnobotany.

***Answer: Thank you for the comment. According to this comment, we have tried to find these points. Due to lack of traditional and ethnobotanical information, we have added only few point. Instead, however, we have included some detailed results of anti-inflammatory activity of this plant that we have published (see L71-78).

Reviewer 4 Report

Manuscript was improved considerably. However, authors should label properly individual figures and immunoblots in Fig.3, there are missing letters next to some graphics. Add statistical differences in figure showing quantitative data for protein expression of p-PLCgama.

Author Response

Reviewer #4

Manuscript was improved considerably. However, authors should label properly individual figures and immunoblots in Fig.3, there are missing letters next to some graphics. Add statistical differences in figure showing quantitative data for protein expression of p-PLCgama.

****Answer: Thank you for the comment. In Fig.3, we found that the label ‘(b)’ of the figure 3b was in the next page. So, we revised the label to be in same page with the figure. And, we have added statistical differences in the quantitative data of p-PLCgamma but there is no significance in extract-treated groups compared to DNP-IgE/DNP-HAS group (see Fig. 3b lower panel, L144-146, and L147).